# Increased Circulating ADMA in Young Male Rats Caused Cognitive Deficits and Increased Intestinal and Hippocampal NLRP3 Inflammasome Expression and Microbiota Composition Alterations: Effects of Resveratrol

**DOI:** 10.3390/ph16060825

**Published:** 2023-05-31

**Authors:** Mei-Hsin Hsu, Yi-Chuan Huang, Yu-Chieh Chen, Jiunn-Ming Sheen, Li-Tung Huang

**Affiliations:** 1Kaohsiung Chang Gung Memorial Hospital, Kaohsiung City 833, Taiwan; a03peggy@yahoo.com.tw (M.-H.H.); gesicht27@gmail.com (Y.-C.C.); 2Chiayi Chang Gung Memorial Hospital, Puzi City 613, Taiwan; ray.sheen@gmail.com

**Keywords:** asymmetric dimethylarginine, endothelium, NLRP3 inflammasome, microbiota, resveratrol, cognition

## Abstract

Endothelial dysfunction is characterized by disturbances in nitric oxide (NO) bioavailability and increased circulating asymmetric dimethylarginine (ADMA) due to the enormous release of free radicals. Increased circulating ADMA may cause endothelial dysfunction and a variety of clinical disorders, such as liver and kidney disease. Young male Sprague-Dawley rats at postnatal day 17 ± 1 received continuous ADMA infusion via an intraperitoneal pump to induce endothelial dysfunction. Four groups of rats (*n* = 10 per group) were allocated: control, control and resveratrol, ADMA infusion, and ADMA infusion and resveratrol groups. Spatial memory, NLR family pyrin-domain-containing 3 (NLRP3) inflammasome, cytokine expression, tight junction proteins in the ileum and dorsal hippocampus, and microbiota composition were examined. We found cognitive deficits; increased NLRP3 inflammasome in the plasma, ileum, and dorsal hippocampus; decreased ileum and dorsal hippocampal cytokine activation and tight junction proteins; and microbiota composition alterations in the ADMA-infusion young male rats. Resveratrol had beneficial effects in this context. In conclusion, we observed NLRP3 inflammasome activation in peripheral and central dysbiosis in young male rats with increased circulating ADMA, and found that resveratrol had beneficial effects. Our work adds to the mounting evidence that inhibiting systemic inflammation is a promising therapeutic avenue for cognition impairment, probably via the gut-brain axis.

## 1. Introduction

Asymmetric dimethylarginine (ADMA) is a competitive inhibitor of nitric oxide synthase (NOS) that can reduce the synthesis of nitric oxide (NO) and promote eNOS uncoupling and superoxide anion production, and is considered a pro-oxidant [1]. Endothelial dysfunction is characterized by disturbances in NO bioavailability and increased circulating ADMA owing to the enormous release of free radicals [2]. The endothelium is directly involved in peripheral vascular disease, stroke, heart disease, diabetes, insulin resistance, chronic kidney failure, aging, smoking, obesity, venous thrombosis, and severe viral infectious diseases [3]. Additionally, increased circulating ADMA levels have been observed in various clinical conditions, including liver disease, renal disease, and multiple organ damage associated with critical illness [1].

Oxidative and nitrosative stress can lead to NLR family pyrin-domain-containing 3 (NLRP3) activation. NLRP3 is a 115 kDa cytosolic protein expressed in monocytes, neutrophils, dendritic cells, and lymphocytes. The activated NLRP3 inflammasome leads to the activation of caspase-1, which mediates the production of interleukin-1β (IL-1β) and interleukin-18 (IL-18) pro-inflammatory cytokines, and the initiation of a rapid form of cell death termed pyroptosis [4]. NLRP3 inflammasome activation may lead to endothelial dysfunction [5]. In addition, NLRP3 increases the levels of proinflammatory factors and accelerates pathological progression in a variety of brain disorders [6].

Inflammasomes are internal sensors of cellular integrity and tissue health. The gut microbiota is considered an internal environmental factor of the human body that may affect host development and many other physiological processes. NLRP3 signaling has been implicated in chronic restraint-stress-induced colonic inflammation and dysbiosis [7]. Accumulating evidence shows that the gut microbiota plays an important role in the development of cognitive impairment, such as Alzheimer’s disease [8].

Resveratrol (3,4′,5-trihydroxy-trans-stilbene) is a natural polyphenolic compound produced by plants in response to environmental stress. Food sources of resveratrol include the skin of raspberries, plums, grapes, blueberries, and mulberries. Resveratrol has various biological properties, including anti-inflammatory, antioxidant, and neuroprotective effects. Resveratrol is known for its beneficial effects on cognition [9]. Animal studies have demonstrated the protective effects of resveratrol in a variety of disease models, including cardiovascular diseases, diabetes, cancer, and neurodegenerative diseases [10]. Resveratrol may act on gut microbiota and exert its beneficial effects on metabolic syndrome [11]. Resveratrol has been found to improve dysbiosis-related metabolic dysregulation in the context of maternal and postnatal high-fat diet exposure [12]. Moreover, resveratrol could alter the expression of the NLRP3 inflammasome, which is critically involved in endothelial dysfunction and oxidative stress, in many organs [10].

We have previously reported that increased circulating ADMA in young male rats caused cognitive deficits, and both melatonin [13] and resveratrol [14] could alleviate the cognitive deficit in this context. Current literature suggests that the NLRP3 inflammasome plays an important role in many brain disorders and could be altered by resveratrol intake. The first aim of this study was to examine whether the NLRP3 inflammasome plays an important role in cognitive impairment in young male rats with increased circulating ADMA, and the potential neuroprotective effects of resveratrol. We also explored the possible alterations in microbiota composition in this context.

## 2. Results

### 2.1. Morris Water Maze

During the water maze tests, all rats learned how to find the platform, and there was no significant difference in swim velocity among the four experimental groups at any time (all *p* > 0.1). Two-way ANOVA revealed that escape latencies improved over time in all four groups, as shown by a significant effect of day (*p* < 0.05), indicating that learning occurred (Figure 1a). There were significant differences among the groups in the number of trial blocks that were required to learn to escape by visual cues (C vs. A, *p* < 0.05, days 1, 2, 3, and 4; A vs. AR, *p* < 0.05, days 1 and 2) (Figure 1a). The above data showed that ADMA infusion caused acquisition memory deficits, which could be prevented by resveratrol. At PND 42, there was a significant difference in retention in the target quadrant between the C and A group (*p* < 0.05). In addition, there was a significant difference in retention time in the target quadrant between the A and AR group (*p* < 0.05). The above results showed that ADMA infusion caused retention memory deficits, which were prevented by resveratrol intake (Figure 1b).

### 2.2. NLRP3 Expression in the Plasma, Dorsal Hippocampus, and Ileum

Rats in group A consistently demonstrated increased NLRP3 expression in the plasma, dorsal hippocampus, and ileum, compared to group C (all *p* < 0.05). The AR group had lower NLRP3 expression than group A (all *p* < 0.05; Figure 2a–c). The above results suggest that ADMA infusion increased NLRP3 inflammasome expression in the plasma, dorsal hippocampus, and ileum. Furthermore, resveratrol treatment reduced NLRP3 inflammasome activation in the peripheral tissue, intestine, and dorsal hippocampus.

### 2.3. Expression of Cytokines in Rat Dorsal Hippocampus

We have previously shown that ADMA infusion in young rats increases circulatory ADMA [13,14]. As ADMA is considered a pro-oxidant, we examined inflammatory cytokines and mediators in the plasma, ileum, and dorsal hippocampus. Rats in group A had higher IL-1α and IL-6 expression in the dorsal hippocampus than those in group C (*p* < 0.05; Figure 3a,c). The AR group exhibited lower IL-6 levels than group A (*p* < 0.05; Figure 3c). Rats in group A had higher IL-1α, IL-6, and TNF-α expression in the ileum than those in group C (*p* < 0.05; Figure 4a,c). The AR group had lower IL-1α and IL-6 expression levels than group A (*p* < 0.05; Figure 4a,c). However, there were no significant differences in IL-1B and IL-18 levels among the four experimental groups in either the dorsal hippocampus or the ileum. The above results suggest that ADMA infusion in young rats increased the expression of inflammatory cytokines in the dorsal hippocampus and ileum, and resveratrol intake could partially reverse this inflammatory status.

### 2.4. Inflammation and Tight Junction Protein in Rat Dorsal Hippocampus and Ileum

Next, we examined the inflammatory mediators and tight junction proteins, including TLR4, Claudin-1, occludin, and ZO-1 by WB. Higher TLR4 and lower Claudin-1, occludin, and ZO-1 expressions in the brain dorsal hippocampus were found in rats in the ADMA group than in rats in the control group (*p* < 0.05; Figure 5a–d). The AR group reversed the decrease in occludin and ZO-1 levels (*p* < 0.05; Figure 4c,d). Rats in group A had higher TLR4 and lower occludin and ZO-1 expression in the ileum than those in group C (*p* < 0.05; Figure 6a–d). However, resveratrol had no beneficial effects on TLR4, occludin, or ZO-1 in the ileum (*p* < 0.05; Figure 6a).

### 2.5. Gut Microbiota Changes in the Context of ADMA Infusion

We explored α- and β-diversity metrics to assess the effects of ADMA infusion on the total gut microbial community. Microbial α-diversity (ACE index) was higher in group A than in group C. The AR groups had lower α-diversity than group A (Figure 7a). We next examined β-diversity using PLS-DA. Scatterplots of PLS-DA analysis are depicted in 1 B. To explore the microbiota responses of ADMA infusion and resveratrol intake, detailed composition alternations of gut microbiota were further visualized at different levels (classes and genera), and the results are shown in Figure 7, Figure 8 and Figure 9. Figure 7b shows significant clustering according to the study group, indicating that the microbial community was distinctly altered by ADMA infusion and resveratrol intake. Figure 7c illustrates the major bacterial classes present in male rats, including *Clostridia*, *Bacteroidia*, *Bacilli*, *Erysipelotrichia*, and *Verrucomicrobiae*. The *Firmicutes*/*Bacteroidetes* (F/B) ratio was lower in group A than in group C. The AR groups had a higher F/B ratio than group A (*p <* 0.05) (Figure 7d). At the class level, *Actinobacteria*, *Bacteroidia*, and *Verrucomicrobiae* abundance was greater in group A than in group C. *Bacteroidia* abundance was decreased more in AR groups than in A groups (Figure 7e) (*p <* 0.05).

Figure 8a illustrates the major bacterial genera present in male rats, including *Anaerotruncus*, *Bifidobacterium*, *Clostridium_sensu_stricto_1* and *Escherichia_Shigella*. At the genus level, the abundance of *Anaerotruncus* was affected by ADMA infusion (Figure 8b). The relative abundance of the genera *Bifidobacterium*, *Clostridium_sensu_stricto_1*, and *Escherichia_Shigella* was significantly increased by ADMA infusion (Figure 8c–e).

We next applied the LEfSe algorithm to identify statistically significant biomarkers among the groups, as depicted in Figure 9. According to LEfSe analysis, a lower abundance of genera *Actinobacteria*, *Bifidobacterium*, *Verrucomicrobiae*, *Clostridium_sensu_stricto_1* and *Bacteroidiabut* and a greater *Anaerotruncus* abundance in the A group than in C group were detected (Figure 8a). Resveratrol intervention resulted in a higher *Anaerotruncus* abundance in the AR group than in the A group, and lower *Bacteroidia* abundance in the A group than in the AR group (Figure 8b).

## 3. Discussion

The principal findings of this study were as follows: (1) increased circulating ADMA in young male rats caused cognitive deficits and increased the NLRP3 inflammasome in the plasma, ileum, and dorsal hippocampus; (2) increased circulating ADMA in young male rats resulted in ileum and dorsal hippocampal cytokine activation and tight junction protein decrease; (3) microbiota composition alterations were encountered in this context; and (4) resveratrol had beneficial effects in alleviating cognitive deficits and NLRP3 inflammasome activation, ameliorating ileum and dorsal hippocampus tight junction protein suppression, and microbiota composition alterations.

The NLRP3 inflammasome is crucial for endothelial homeostasis [15]. In this study, we expanded on our previous work [14] and showed that increased circulating ADMA in young male rats causes an increase in the NLRP3 inflammasome in the plasma, ileum, and dorsal hippocampus. We also found that resveratrol could reduce the NLRP3 inflammasome increase in the ileum and dorsal hippocampus. Zou et al. showed that the activation of the NLRP3 inflammasome and subsequent inflammatory responses in the cerebral cortex are involved in the process of traumatic brain injury. Resveratrol may attenuate the inflammatory response and relieve traumatic brain injury by reducing reactive oxidant species production and inhibiting NLRP3 inflammasome activation [16]. Resveratrol alone has been shown to induce autophagy and evoke anti-inflammatory responses in retinal pigment epithelium cells, as well as inhibit NLRP3 inflammasome expression, caspase-1 activation, and IL-1β secretion in a mouse model of acute lung injury [17]. Dong et al. showed that resveratrol induced a myocardial protective effect due to its relationship with NLRP3 inflammation [18]. This study adds new insights to our previous study by showing that resveratrol is effective in reducing peripheral and central NLRP3 inflammasome activation in young male rats with increased circulating ADMA levels.

We have previously reported that plasma ADMA levels were increased in young male rats receiving continuous ADMA infusion for 4 weeks [13]. In this study, we found that increased circulating ADMA in young male rats leads to NLRP3 inflammasome activation and enhancement of pro-inflammatory cytokines. We also showed that increased circulating ADMA in young male rats could damage the tight junction proteins in the ileum and dorsal hippocampus, and its mechanisms may be related to TLR4/caspase 1 and 3, NLRP3 alterations in the ileum and dorsal hippocampus, and dysbiosis. Tight junction proteins, such as Claudin-1, occludin and ZO-1 in the endothelium are important for maintaining intestinal and blood–brain barrier (BBB) integrity [19]. In this study, we found that tight junction proteins were differentially affected by in ileum and dorsal hippocampus. In dorsal hippocampus, we found decreased Claudin-1, Occludin 1, and ZO-1. Resveratrol intake increased the expression of Occludin 1 and ZO-1. In ileum, we found decreased Occludin 1 and ZO-1. However, resveratrol intake has little impact on the expression of Occludin 1 and ZO-1. The above findings suggest that NLRP3 inflammasome caused different insults in the intestine barrier and the BBB. Our finding in this study supports the recently proposed concept that peripheral inflammation may result in cognitive impairment [20].

The altered α and β diversities in this study broadly parallel other conditions linked with microbiota composition alterations in brain disorders such as Parkinson’s disease, multiple sclerosis, Alzheimer’s disease, and myasthenia gravis [21,22,23,24]. A lower F/B ratio has been found to be associated with a pro-inflammatory shift of the gut microbiota in brain disorders such as multiple sclerosis, Alzheimer’s disease, and myasthenia gravis [22,23,24]. In this study, we found that increased circulating ADMA in young male rats decreased the F/B ratio, suggesting a pro-inflammatory shift.

LEfSe analysis further confirmed that most intestinal microbes belonging to *Bacteroidota* decreased and those belonging to *Firmicutes* increased in group A rats. The harmful effects of ADMA may be due to alterations in the gut microbiota. We observed that ADMA increased class *Bacteroidia* and resveratrol could reduce it to the level of the C group. Increased class *Bacteroidia* has been linked to chronic social defeat stress, presumably because of its inflammation effect [25]. We also observed that ADMA decreased genus *Anaerotruncus* and resveratrol could restore it to the level of the C group. Diet-induced increase in plasma trimethylamine-N-oxide, a toxin for endothelium produced from diet by phylum *Bacteroidets*, is linked with a decrease in genus *Anaerotruncus* [26]. A decreased genus *Anaerotruncus* has been found in AD patients [27].

The harmful effects of ADMA can be attributed to dysbiosis. Increased ADMA may result in oxidative and nitrosative stress in vascular endothelial and smooth muscle cells [28,29]. Oxidative and nitrosative stress can lead to higher expression of pro-inflammatory cytokines and activate NLRP3 [30,31]. Inflammatory responses, metabolic changes, and dysbiosis are intimately related to intestinal barrier disruption [7]. The protective mechanisms of resveratrol may be due to alterations in the gut microbiota. Owing to its low bioavailability after oral intake, it is conceivable that resveratrol might reach the intestine and modulate microbiota compositions [32].

We observed that resveratrol increased the abundance of Bacteroidia (at the genus level). It is known that the gut microbiota actively participates in resveratrol metabolism by increasing its bioavailability from resveratrol precursors and producing different resveratrol derivatives [11].

Shen et al. transplanted the gut microbiota from patients with Alzheimer’s disease (AD) into APP/PS1 double transgenic mice and found activation of the NLRP3 inflammasome in the intestinal tract of mice, subsequently causing the release of inflammatory factors and more severe cognitive impairment [33]. They reported that NLRP3 inflammasome could be activated in AD patients and may be a therapeutic target of AD-related neuroinflammation [33]. Similarly, we found that increased circulating ADMA in young male rats changed the microbiota composition and intestinal NLRP3 activation, which were associated with dorsal hippocampal NLRP3 activation and cognition impairment. Together, the evidence suggests that targeting dysbiosis may be a promising way to alleviate neuroinflammation.

Few studies have attempted to address the role of gut microbiota composition in young animals with endothelial dysfunction. However, this study had several limitations. First, we examined all the parameters at one time; therefore, we could not determine the causal relationship between NLRP3 inflammasome activation, microbiota composition alterations, and spatial deficits. Further work, including animal experiments and longitudinal human studies, will be needed to determine the causal relationship between NLRP3 inflammasome activation and gut microbiota composition alterations and the pathogenesis of endothelium dysfunction in adolescents and young adults. Second, only male rats were used in the current study; thus, further studies are warranted to characterize the outcomes across sexes. Finally, we should note that these findings were obtained in a rodent study, and translation of the findings to human physiology and disease should be cautioned.

## 4. Materials and Methods

### 4.1. Animals

The experiments were performed according to the Guidelines for Animal Experiments of the Chang Gung Memorial Hospital and Chang Gung University. Sprague–Dawley rats were used in this experiment, and the day of birth was designated as postnatal day 0 (PND 0). Male Sprague-Dawley rats (PND 17 ± 1) weighing ~50 g were used. All animals were housed in a room maintained at 24 °C with 12-h light/dark cycles and had free access to standard food.

### 4.2. Animals and Grouping

All animals were implanted with intraperitoneal (i.p.) Alzet pumps and allocated into four experimental groups (PND 17). The animal numbers were determined based upon our previous work [13,14]. Four groups of rats (*n* = 10 per group) were allocated. Control rats (group C; *n* = 10) received saline injections. The second group of rats received saline + resveratrol (group CR; *n* = 10). Resveratrol was administered in a dose of 50 mg/kg/day in drinking water, a dose similar to our previous study. The third group received ADMA (0.25 mmol/kg/day) [13,34] (group A; *n* = 10). The fourth group received an injection of ADMA and oral resveratrol (group AR; *n* = 10). All rats were sacrificed on PND 45 for further molecular and biochemical studies. To avoid gender effects in puberty stage, we chose to use only male rats in this study.

### 4.3. Tissue Collection

The rats were euthanized at PND 45 after behavioral studies were performed, and the blood plasma, brain dorsal hippocampus and ileum were immediately collected for the study of microbiota composition.

### 4.4. Morris Water Maze Test

The Morris water maze test was conducted to assess hippocampus-dependent spatial learning and memory in all rats between ~PND 39 and ~PND 44, as previously reported [13]. Latency to reach the platform, distance traveled, and average swimming speed were recorded between ~PND 40 and ~PND 43 (learning phase). Retention of memory was performed with the platform absent from the pool at ~PND 44 (retention memory test).

### 4.5. Enzyme-Linked Immunosorbent Assay (ELISA)

Commercial ELISA kits were used to measure plasma NLRP3 (MyBioSource *Cat. No* MBS1600620, San Diego, CA, USA), according to the manufacturer’s protocol.

### 4.6. Western Blot

Western blot analysis was performed on the dorsal hippocampus and intestinal tissue samples. Briefly, proteins of the dorsal hippocampus and intestines were isolated, separated by electrophoresis, transferred to a PVDF membrane, and probed with the following primary antibodies: NLRP3 (1:1000, A12694, ABclonal, Woburn, MA, USA), caspase 1 (1:1000, ZRB1233, Sigma-Aldrich, Burlington, MA, USA), caspase 3 (1:1000, sc-56053, Santa Cruz Biotechnology, Dallas, TX, USA), TLR4 (1:1000, SC-293072, Santa Cruz Biotechnology, Dallas, TX, USA), claudin-1 (1:1000, ab15098, Cambridge, MA, USA), occludin (1:1000, SC-8144, Santa Cruz Biotechnology, Dallas, TX, USA), and ZO-1 (1:1000, ab96587, Cambridge, MA, USA). The integrated optical density (IOD) was factored into Ponceau S staining to correct for any variations in total protein loading, as we previously reported [13]. Bands of interest were visualized using electrochemiluminescence reagents (PerkinElmer, Waltham, MA, USA) and quantified by densitometry (Quantity One Analysis software; Bio-Rad, Hercules, CA, USA) as the IOD after subtraction of the background. The protein abundance was represented as IOD/Ponceau S.

### 4.7. Detection of Cytokines in Tissue Protein

Tissue protein was obtained from the dorsal hippocampi, kidneys, and intestines of SD rats at a concentration of 2 g/kg protein. IL-1α, IL-1β, IL-6, IL-18, and TNF-α were detected using a Milliplex MAP Rat Cytokine/Chemokine Magnetic Bead Panel kit (Cat. RECYTMAG65K; EMD Millipore, Darmstadt, Germany).

### 4.8. Gut Microbiota Profiling in the Context of ADMA Infusion

Feces (100 mg) from the four experimental groups were collected and treated as previously reported [12]. All PCR amplicons were mixed and sent to Biotools Co., Ltd. (Taipei, Taiwan) for sequencing using an Illumina Miseq platform (Illumina, San Diego, CA, USA). Genomic DNA was subjected to amplification of the V3–V4 hypervariable regions of the 16D rRNA gene, as we previously reported [12]. Raw tags were obtained using the FLASH software. Raw sequence data were processed in QIIME version 1.9.1 to obtain high-quality, clean tags. Quality filtering and ambiguous base removal were performed. Chimeric sequences were removed using the UCHIME algorithm. Sequences with ≥97% similarity were clustered into the same operational taxonomic units (OTUs) using the UCLUST algorithm. The medians of 91,024 raw sequencing reads and 74,576 effective tag sequences per sample were obtained. The average efficacy rate was 82%. Phylogenetic relationships were constructed with FastTree based on a representative sequence alignment. We investigated the diversity patterns of the microbial communities. *α*-diversity was measured using the abundance-based coverage estimator (ACE) index. We assessed the β-diversity of the gut microbiota across groups using partial least squares discriminant analysis (PLS-DA). Linear discriminant analysis effect size (LEfSe) was applied to identify the biomarker taxa of each group.

### 4.9. Statistical Analysis

We assumed that resveratrol might reduce circulating-ADMA-induced biological harms peripherally and centrally. Western blotting, ELISA, and cytokine levels were analyzed by one-way ANOVA with Bonferroni post hoc test. Results from the Morris water maze acquisition memory were evaluated using a two-way ANOVA (grouped as between subjects) with repeated measures (day), followed by LSD post hoc tests. Microbiota composition was analysed as previously described [12]. Alpha diversity (number of OTUs, species evenness, and Shannon’s diversity index) was analysed using Calypso. To identify OTUs that were differentially abundant between the groups, we used two different statistical analyses. Linear Discriminant Analysis (LDA) Effect Size (LEfSe) was performed using the Galaxy web application (60) using default settings. The R package Phyloseq was used for the negative binomial Wald test in DESeq2. LEfSe was performed across all taxonomic levels (phylum to OTU level), whereas DESeq2 was performed at the OTU level. Distance-based linear model (DistLM) analysis was performed using PRIMER to examine associations between the gut microbiota and metabolic parameters. All analyses were performed using the Statistical Package for the Social Sciences (SPSS) software (VERSION 15) on a PC-compatible computer. For all variables measured, outliers that lay 1.5 interquartile ranges (IQRs) below the first quartile or 1.5 IQRs above the third quartile were removed from the analysis. Values are expressed as mean ± SEM, and significance was defined as *p* < 0.05.

## 5. Conclusions

In this study, we observed correlational changes of NLRP3 inflammasome activation, endothelium dysfunction, and microbiota composition changes in young male rats with increased circulating ADMA, and resveratrol had beneficial effects in this context. Our work adds to the mounting evidence that inhibiting systemic inflammation is a promising therapeutic avenue for cognitive impairment. Since our findings in this study are largely correlational, future works should take advantage of longitudinal studies to determine the progression of the observed effects, or mechanistic studies to investigate the cause-and-effect relationship between ADMA, inflammation, and cognitive performance.

## Figures and Tables

**Figure 1 pharmaceuticals-16-00825-f001:**
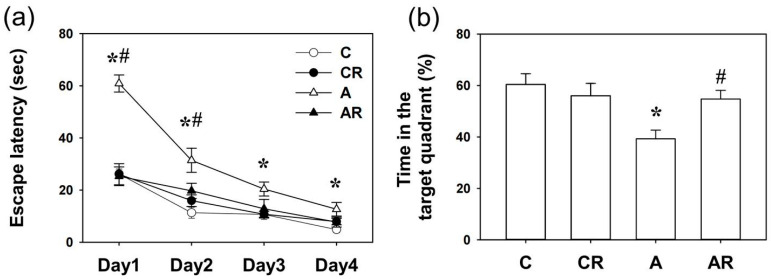
Spatial memory assessed by the Morris water maze. (**a**) Escape latencies to the invisible platform in the Morris water maze test (mean ± SEM). Rats in the A group swam for a longer period to locate the invisible platform on acquisition days 1, 2, 3, and 4 than rats in the other three groups (all *p* < 0.05). Resveratrol intake reversed this deficit (A vs. AR, *p* < 0.05, on days 1 and 2 of acquisition). (**b**) A significant difference in retention memory was detected between the four experimental groups on day 5. Group A exhibited less retention time in the previously invisible platform quadrant than group C (C vs. A, *p* < 0.05). Resveratrol intake reversed this deficit (A vs. AR, *p* < 0.05). * *p* < 0.05, vs. C group; # *p* < 0.05 vs. A group. C: control group; CR: control rats that received oral resveratrol; A: asymmetric dimethylarginine infusion group; AR: asymmetric dimethylarginine infusion group that received oral resveratrol; *n* = 10, each group.

**Figure 2 pharmaceuticals-16-00825-f002:**
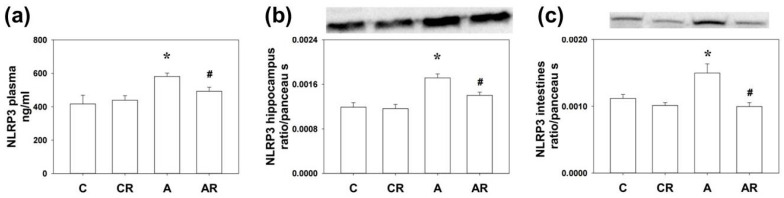
NLRP3 inflammasome activation in plasma (**a**), dorsal hippocampus (**b**), and ileum (**c**). Increased NLRP3 expression in group A rats was consistently found in the plasma (ELISA), dorsal hippocampus (WB), and ileum (WB) as compared to group C. The AR group had lower NLRP3 expression than group A. * *p* < 0.05 vs. C group; # *p* < 0.05 vs. A group. *n* = 7–10, each group.

**Figure 3 pharmaceuticals-16-00825-f003:**
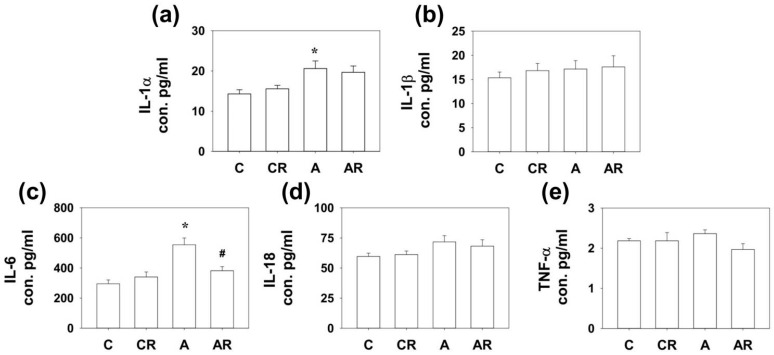
Cytokine expression in dorsal hippocampus. Increased IL-1α (**a**) and IL-6 (**c**) expression was found in the dorsal hippocampus of rats in group A compared to those in group C. In addition, the AR group had lower IL-6 (**c**) expression than group A. However, there were no significant differences in IL-1β (**b**), IL-18 (**d**) and TNF-a (**e**) among the four study groups. * *p* < 0.05 vs. the C group; # *p* < 0.05 vs. A group. *n* = 7–10, each group.

**Figure 4 pharmaceuticals-16-00825-f004:**
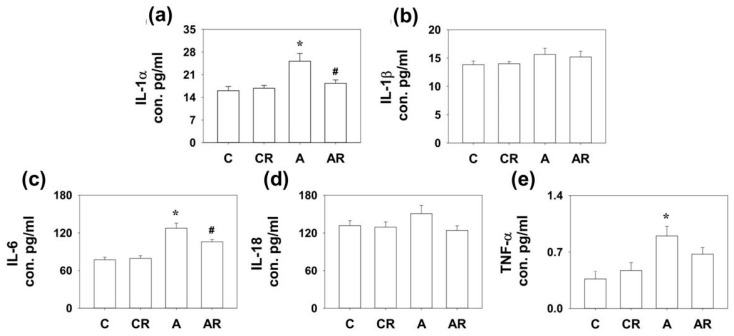
Cytokine expression in ileum. Increased IL-1α (**a**), IL-6 (**c**), and TNF-α (**e**) expression was detected in the dorsal hippocampus of rats in group A than those in group C. The AR group had lower IL-1α (**a**) and IL-6 (**c**) expression than group A. However, there were no significant differences in IL-1β (**b**) and IL-18 (**d**) among the four experimental groups. * *p* < 0.05 vs. the C group; # *p* < 0.05 vs. A group. *n* = 7–10, each group.

**Figure 5 pharmaceuticals-16-00825-f005:**
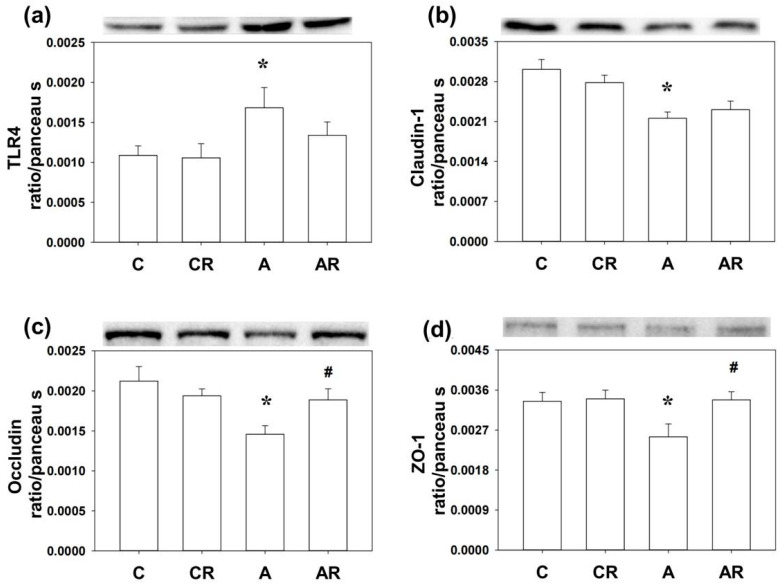
Western blot results in the dorsal hippocampus. Increased TLR4 (**a**) and decreased Claudin-1 (**b**), occludin (**c**), and ZO-1 (**d**) expression were found in group A compared to group C. The AR group reversed the decrease of occludin (**c**) and ZO-1 (**d**) levels in group A. * *p* < 0.05 vs. the C group; # *p* < 0.05 vs. the A group.

**Figure 6 pharmaceuticals-16-00825-f006:**
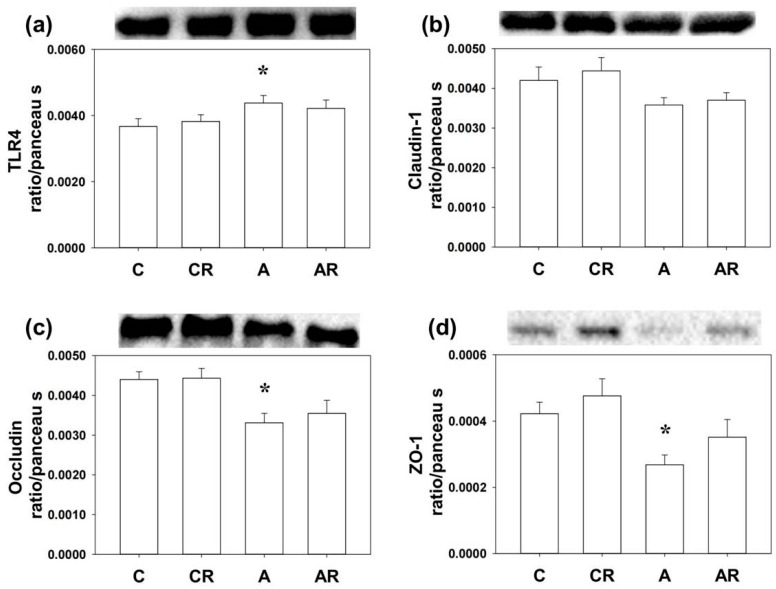
Western blot results in the ileum. Increased TLR4 (**a**) and the decrease of occludin (**c**) and ZO-1 (**d**) expression were found in the ileum of rats in group A as compared with group C. However, Claudin-1 (**b**) and resveratrol had no beneficial effects in the ileum of rats infused with ADMA. * *p* < 0.05 vs. the C group.

**Figure 7 pharmaceuticals-16-00825-f007:**
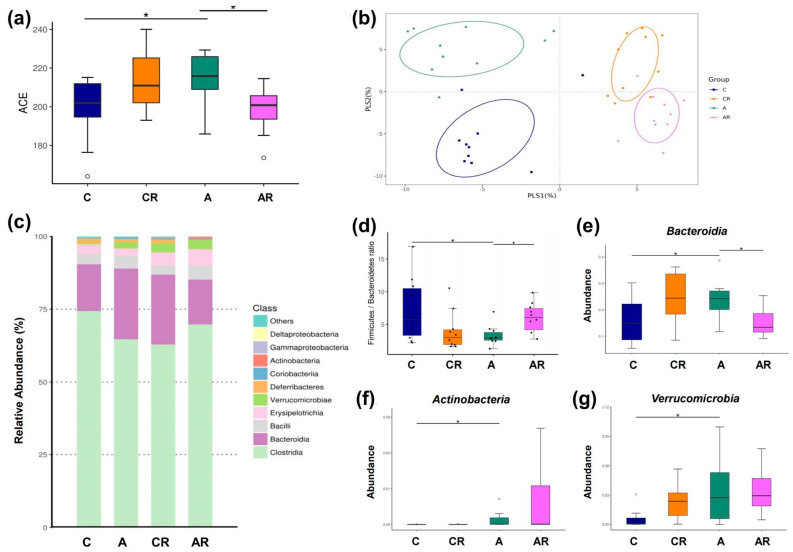
Gut microbiota composition in four experimental groups. α-diversity measured by abundance-based coverage estimator (ACE) index (**a**). β-diversity using partial least squares discriminant analysis (PLS-DA) (**b**). Relative abundance of the top 10 classes of gut microbiota (**c**). *Firmicutes* to *Bacteroidetes* (F/B) ratio (**d**). Relative abundance of the class *Bacteroidia* (**e**), *Actinobacteria* (**f**), *Verrucomicrobiae* (**g**). Data are shown as means ± SEM; *n* = 10/group. * *p <* 0.05.

**Figure 8 pharmaceuticals-16-00825-f008:**
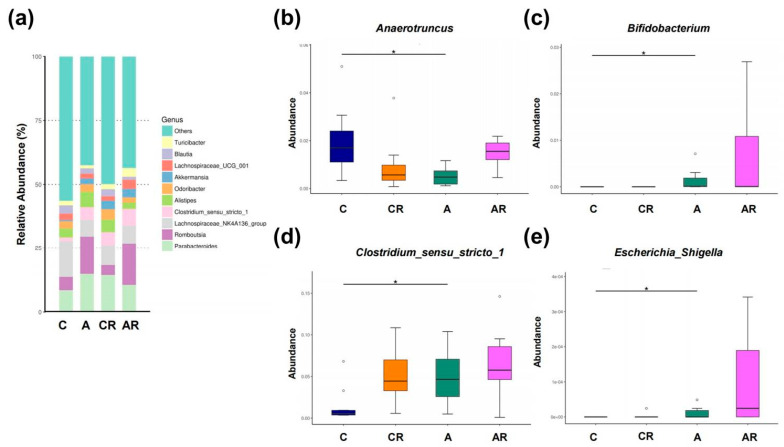
Gut microbiota composition in genera.Relative abundance of the top 10 genera in the gut microbiota (**a**). Relative abundances of the genera *Anaerotruncus* (**b**), *Bifidobacterium* (**c**), *Clostridium_sensu_stricto_1* (**d**), *Escherichia_Shigella* (**e**). Data are shown as means ± SEM; *n* = 10, each group. * *p <* 0.05.

**Figure 9 pharmaceuticals-16-00825-f009:**
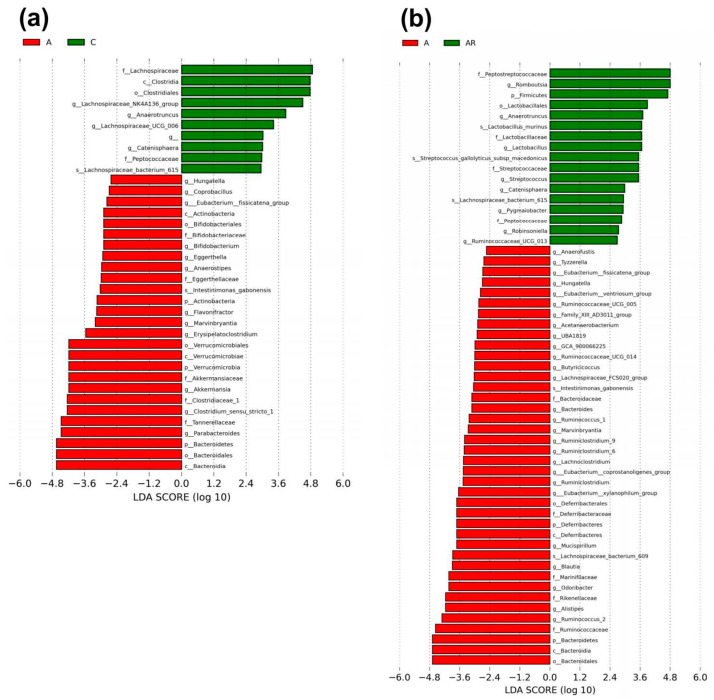
Gut microbiota composition in LEfSe. Linear discriminant analysis effect size (LEfSe) was used for biomarker discovery. Most depleted and enriched bacterial taxa in the (**a**) A (red) vs. C (green) group and (**b**) A (red) vs. AR (green) group are shown. The threshold for the linear discriminant was set to 2. *n* = 10, each group.

## Data Availability

Data is contained within the article.

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
