# Peer review of "Increased Circulating ADMA in Young Male Rats Caused Cognitive Deficits and Increased Intestinal and Hippocampal NLRP3 Inflammasome Expression and Microbiota Composition Alterations: Effects of Resveratrol"

_pharmaceuticals, 2023, doi:10.3390/ph16060825_

Round 1

Reviewer 1 Report

This is an excellent study in young male rats in which asymmetric dimethylarginine (ADMA) (or saline control) was administered by i.p. Alzet minipumps, and animals received the natural product antioxidant resveratrol (R) (or not) in drinking water. The animals were tested during development for impairment of memory (Morris water maze) and sacrificed at ~7 weeks of postnatal age for assay of plasma, dorsal hippocampus, and ileal levels of cytokines, NLRP3 inflammasomes, and endothelial tight junction proteins. Feces were examined for populations of bacteria in their microbiomes. The authors report that ADMA peritoneal infusion increased NLRP3 levels (activation?)and cytokine levels in all tissues, reduced tight junction proteins in hippocampus and ileum, and altered microbiome bacterial populations in feces. All of these effects of ADMA were reversed by resveratrol.

The authors thus established a correlative link between inhibition of NOS by ADMA, impaired cognition (memory) and activation of inflammation, with a strong hint that the microbiome may mediate this effect.

I find no criticisms of the methods or conclusions.  I do find that the paper would be enhanced by: 1. indication of the plasma levels of ADMA achieved by their approach (they may already have reported these data in earlier studies), and 2. more discussion about how resveratrol might be working, and how these potential mechanisms might be approached (e.g., assay of oxidative stress damage to proteins, etc).

But these are minor concerns. Overall I support publication of this paper in its current form.

No changes needed.

Reviewer 2 Report

This study presents intriguing data that contribute to our understanding of the potential effects of Asymmetric Dimethylarginine (ADMA) infusion and resveratrol treatment on cognitive abilities, inflammatory responses, tight junction protein expression, and gut microbiota composition in rats. Overall, the introduction and methods sections of your manuscript are detailed, well-structured, and the research hypothesis is clearly outlined. However, there are a few suggestions for improvement. These comments are meant to enhance the clarity and overall quality of your manuscript. The results are generally well presented, with statistical analysis appropriately applied. The discussion integrates the results with existing literature, and potential mechanisms are proposed. Nevertheless, there are some areas that would benefit from further clarification and discussion.

Introduction:

Line 38: I suggest rewording to "Additionally, increased circulating ADMA levels have been observed in various clinical conditions..." to make it more concise.

Line 44: Consider clarifying what you mean by "enzyme uncoupling". You might want to explain briefly how this contributes to oxidative stress.

Line 66: The transition between the paragraphs is abrupt. I would recommend adding a sentence that links the role of NLRP3 in endothelial dysfunction and oxidative stress to the potential therapeutic role of resveratrol.

Line 72: The transition to the study aims is a bit abrupt. Consider introducing the aims with a sentence that briefly summarizes the current gaps in the literature that your study addresses.

Materials and methods:

Line 85: It's not clear why you used male rats specifically. If there is a reason related to the study design or the model of disease, please explain.

Line 89: Please clarify how you determined the "minimum number of animals".

Line 96: There is a repetitive subtitle, "Tissue collection", which appears twice (lines 90 and 96). I assume the second one should be "Behavioural Testing" or "Morris Water Maze Test".

Lines 125-142: The method for microbiota analysis is well-described, but it may be beneficial to explain why you selected the specific regions of the 16D rRNA gene for amplification.

Line 143-160: The statistical analysis section is clear, but please consider providing more information about the assumptions of the tests used, and how you tested for these assumptions.

Line 158: It is not clear why you chose to remove outliers that are 1.5 IQRs below the first quartile or above the third quartile. Was this an arbitrary decision, or is there a specific reason for this criterion? If so, please clarify.  

Results: 

The results are generally well presented and clear. The statistical tests used are appropriate and the authors have made efforts to interpret the results appropriately. However, the authors could further clarify the groups used in the study for readers not familiar with the experiment's design.

The authors appropriately interpret their results within the context of their experiment. They link their findings to existing literature and propose potential mechanisms for their observations. However, the conclusions drawn are largely correlational. The authors should make it clear that while they observed these effects in their model, the exact causative mechanisms remain speculative.

Discussion:

The discussion offers a thorough review of the results within the context of the broader scientific literature. The authors adequately discuss potential mechanisms and implications of their findings. However, the discussion could benefit from an expanded comparison with similar studies, discussing any contradictions or consistencies in findings.

The authors acknowledge several limitations of their study, including the inability to determine a causal relationship due to the experimental design. It's appreciated that they propose the need for future research to address these limitations. However, the authors should also discuss the potential limitations of their rat model and any implications this might have on the translation of their findings to human physiology and disease.

While the authors acknowledge the need for future work, more specific suggestions for future studies would strengthen the paper. For instance, longitudinal studies to determine the progression of the observed effects, or mechanistic studies to investigate the cause-and-effect relationship between ADMA, inflammation, and cognitive performance could be proposed.

Minor editing of English language required

Reviewer 3 Report

The approach is interesting to highlight that "inhibiting systemic inflammation is a promising therapeutic avenue for cognitive impairment"; but the authors should improve the experimental design about the normalization and quality of western blot results: a specific and constitutively expressed protein, that does not depend on the treatments performed, should be used to normalize the different proteins investigated.

Moreover, the quality of ECL signal is poor: sometimes, the signal is barely visible and grainy while it is often saturated.

Why the authors used two different methods (ELISA and Western blot) for the analysis of NLRP3 expression? The first one is a quantitative method, the WB is not.

The descriptions of the legends are sometimes difficult to understand. It is often necessary to check the test to understand the figures.

The authors do not comment on their results obtained on the expression of tight junction proteins, such as Claudin-1, Occludin and ZO-1; they  reduce the discussion to the importance of these proteins for maintaining the integrity of the intestinal and blood-brain barrier...

In fig 6, Claudin does not seem to be modulated by the treatment with A, while the other proteins are affected by A.

AR does not modify the effect of A in any of these ileal tight junction proteins; while  AR only modulates Occludin in  the hippocampus  and ZO-1, Claudin-1 and TLR4 are not reversed (fig. 5).

Have the authors a hypothesis about these results?

Furthermore, why the proteins, that are not modulated in the experimental model, are also highlighted in the figure? It would be enough to describe the result of non-modulation in the text; everything would be more understandable

Minor revision:

in line 43: (NLRP3) is doulbe...

in line 244: the decrease of occluding (C)

Please, check the text to differentiate the numbering of the figures from the abbreviations, for example, "fig. (A) and A= asymmetric dimethylarginine infusion group"; it is confusing. 

You should also verify the correctness of punctuation.

 Moderate editing of English language is required for spelling and punctuation

Round 2

Reviewer 3 Report

I read the author's answeres, now the manuscript is more understandable

However, even if the method used to normalize the signals in W.B. it is acceptable, I personally think it is more correct and useful to use a constitutive protein as a normalization method. Also because some signals are quite saturated,  others are smeared and the entire blot is not visible. 

Minor revision

Line 132: Feces (100 mg) from the four experimental groups were collected and treated as 133 previously reported [17].

The authors eliminated reference [17], but in this revised version, what is the new reference?
